# Developing Feasible, Locally Appropriate Socioeconomic Support for TB-Affected Households in Nepal

**DOI:** 10.3390/tropicalmed5020098

**Published:** 2020-06-10

**Authors:** Bhola Rai, Kritika Dixit, Tara Prasad Aryal, Gokul Mishra, Noemia Teixeira de Siqueira-Filha, Puskar Raj Paudel, Jens W. Levy, Job van Rest, Suman Chandra Gurung, Raghu Dhital, Knut Lönnroth, S Bertel Squire, Maxine Caws, Tom Wingfield

**Affiliations:** 1Birat Nepal Medical Trust, Kathmandu 44600, Nepal; bhola@bnmt.org.np (B.R.); kritika@bnmt.org.np (K.D.); tara@bnmt.org.np (T.P.A.); gokulmishra@gmail.com (G.M.); scgurung@bnmt.org.np (S.C.G.); raghu.dhital@bnmt.org.np (R.D.); maxine.caws@lstmed.ac.uk (M.C.); 2Social medicine, Infectious diseases, and Migration (SIM) Group, Department of Public Health Sciences, Karolinska Institutet, 10653 Stockholm, Sweden; knut.lonnroth@ki.se; 3LIV-TB Collaboration, Departments of International Public Health and Clinical Sciences, Liverpool School of Tropical Medicine, Liverpool L35QA, UK; Noemia.TeixeiradeSiqueiraFilha@lstmed.ac.uk (N.T.d.S.-F.); bertie.squire@lstmed.ac.uk (S.B.S.); 4KNCV Tuberculosis Foundation, 2514 JD The Hague, The Netherlands; puskar.paudel@kncvtbc.org (P.R.P.); jens.levy@kncvtbc.org (J.W.L.); job.vanrest@kncvtbc.org (J.v.R.); 5Tropical and Infectious Disease Unit, Liverpool University Hospitals NHS Foundation Trust, Liverpool L35QA, UK

**Keywords:** tuberculosis, poverty, catastrophic costs, socioeconomic support, social protection, Nepal

## Abstract

Tuberculosis (TB), the leading single infectious diseases killer globally, is driven by poverty. Conversely, having TB worsens impoverishment. During TB illness, lost income and out-of-pocket costs can become “catastrophic”, leading patients to abandon treatment, develop drug-resistance, and die. WHO’s 2015 End TB Strategy recommends eliminating catastrophic costs and providing socioeconomic support for TB-affected people. However, there is negligible evidence to guide the design and implementation of such socioeconomic support, especially in low-income, TB-endemic countries. A national, multi-sectoral workshop was held in Kathmandu, Nepal, on the 11th and 12th September 2019, to develop a shortlist of feasible, locally appropriate socioeconomic support interventions for TB-affected households in Nepal, a low-income country with significant TB burden. The workshop brought together key stakeholders in Nepal including from the Ministry of Health and Population, Department of Health Services, Provincial Health Directorate, Health Offices, National TB Program (NTP); and TB/Leprosy Officers, healthcare workers, community health volunteers, TB-affected people, and external development partners (EDP). During the workshop, participants reviewed current Nepal NTP data and strategy, discussed the preliminary results of a mixed-methods study of the socioeconomic determinants and consequences of TB in Nepal, described existing and potential socioeconomic interventions for TB-affected households in Nepal, and selected the most promising interventions for future randomized controlled trial evaluations in Nepal. This report describes the activities, outcomes, and recommendations from the workshop.

## 1. Introduction

Tuberculosis (TB), the archetypal disease of poverty, kills 1.5 million people each year [1] and typifies health inequity [2,3,4,5,6]. Not only do poorer people have higher likelihood of TB exposure, infection and disease, but they are also more likely to have difficulties in accessing TB diagnosis and care, and to become poorer due to their illness (the “medical poverty trap”) [6,7,8]. Among the reasons behind this are stigma (both TB- and poverty-related), hidden costs of “free” TB diagnosis and care, and a lack of access to social protection (e.g., health and sickness insurance) [9,10], all of which hamper the ability of TB-affected households to access and engage with TB services [11,12]. This can lead to catastrophic TB-related costs and compound impoverishment, especially in the poorest TB-affected households of TB endemic low- and middle-income countries (LMICs) [13,14,15,16]. 

The Sustainable Development Goals (SDGs) recognise that health, poverty, and wellbeing are inextricably linked and must all be addressed in concert [17]. Aligned with the SDGs, the World Health Organization’s (WHO) 2015 End TB Strategy [18] includes a target of “zero TB-affected families facing catastrophic costs” by 2020. However, national TB patient costs surveys have been performed in fewer than 20 countries to date and the costs estimated from these surveys suggest that we are not on course to meet WHO’s catastrophic costs target [19]. To eliminate TB-affected household costs, the End TB Strategy advocates provision of social and economic (socioeconomic) support for TB-affected households [18]. Socioeconomic support could be achieved through long-term national social protection schemes (social security protection and guarantees aimed at preventing or alleviating poverty, vulnerability, and social exclusion) [20] and/or other socioeconomic interventions for TB-affected households including but not limited to cash transfers, mutual support groups, or nutritional support, targeted especially towards the poorest and most vulnerable [21,22]. There is limited evidence to guide the implementation of the End TB Strategy policy change regarding socioeconomic support and it remains unclear how such support can be delivered to TB-affected households in practical, achievable ways, especially in LMICs.

To address this knowledge gap in Nepal, a low-income TB-endemic country, a mixed-methods study was conducted, which integrated: (i) a cohort study of people with and without TB to evaluate the socioeconomic impact of TB in Nepal; (ii) a qualitative study gathering opinions of key stakeholders on the barriers and facilitators to accessing and engaging with TB diagnosis and care in Nepal; and (iii) a national, multi-sectoral workshop with the same key stakeholders to review the preliminary results of the cohort and qualitative studies and to create a shortlist of feasible, locally-appropriate socioeconomic interventions for TB-affected households. The final results of the cohort and qualitative study will be reported elsewhere. Here, we report the activities, outcomes, and recommendations from the national workshop.

## 2. Workshop Overview

A national workshop, the first in Nepal to specifically address socioeconomic support for TB-affected households, was conducted over two days on the 11th and 12th September 2019 in Kathmandu, Nepal. Sixty-five national-, provincial-, and community-level stakeholder participants from multiple sectors participated in the workshop including: the Ministry of Health and Population (MoHP), Department of Health Services, National Tuberculosis Control Center (NTCC), Provincial Health Directorate, Health Offices, and partner organizations working in TB (including non-governmental and civil-society organizations); and also TB/Leprosy Officers, community leaders, female community health volunteers (FCHVs), and people with TB.

Prior to the workshop, we conducted a review of the existing global evidence on socioeconomic support for TB-affected people and the ongoing studies addressing the socioeconomic impact and consequences of TB in Nepal. This review fed into the workshop itself, which was divided into two complementary sections (see Appendix A for agenda and group work materials). Alongside presentations from the MoHP and NTC, Section 1 of the workshop consisted of presentations summarizing our review and preliminary findings of ongoing studies in Nepal, which were then discussed and debated among participants. Section 2 of the workshop focused on multi-disciplinary small group work to: (i) identify the existing psychosocial and economic support available for TB-affected households in Nepal, suggest refinements to these packages, and/or identify elements that could contribute to new support packages; (ii) consider targeting of the intervention (e.g., who should receive support) and potential funding sources; and (iii) design and vote anonymously to select the most feasible, acceptable, and locally appropriate designs for interventions to evaluate during a future randomized controlled trial.

### 2.1. A Review of the Existing Evidence on Socioconomic Support for TB-Affected People

In many resource-constrained settings, including the Indian sub-continent and Nepal, out-of-pocket payments constitute the majority of total health expenditure [23]. These payments put an enormous economic strain on patients and their households, leading to a substantial proportion incurring catastrophic health expenditure. Where available, social protection can defray such out-of-pocket expenses and mitigate catastrophic health expenditure, especially for poverty-related diseases such as TB, which are also associated with profound loss of income [9,10,24]. However, despite social protection coverage continuing to expand, only one quarter of people worldwide have adequate social protection cover and more than half of people have no social protection coverage whatsoever [24]. This dearth of social protection is especially concentrated in low-income countries, in which less than 20% of people are covered [25]. 

It has long been recognised that social protection has the potential to contribute to TB control and elimination [7]. With regards to TB-specific interventions (e.g., those focused only towards people affected by TB), randomised controlled trials, systematic reviews and meta-analyses have shown that TB treatment outcomes can be improved by cash transfers [26] and psycho-emotional and/or socioeconomic support packages [27,28,29]. Other studies have shown that regional or national social protection programmes can be “TB-sensitive” or “TB-inclusive” (e.g., support TB-affected people although not being primarily directed towards them) and contribute to improving TB treatment outcomes [30,31,32] and reducing TB incidence, prevalence, and mortality [9,10]. Indeed, social protection and socioeconomic support is not only included as part of a key pillar of the End TB Strategy, but also in the WHO’s clinical TB treatment guidelines [33]. 

Nepal is a lower-income country with a TB case notification rate of 151 per 100,000 people [34]. In Nepal, 30% of TB cases are missed (e.g., not diagnosed, notified, or treated) and TB is associated with significant mortality, being the seventh leading cause of death [34]. WHO and World Bank data suggest that less than half the population is reached by a basic social protection floor and one-quarter lives below the poverty line [35]. Despite established directly observed therapy (DOT), expansion of GeneXpert molecular diagnostic capabilities, and increased active case finding activities, delivery of TB care in Nepal is hampered by profound geographical challenges, including mountainous areas with poor road access, which can negatively impact TB treatment and prevention outcomes [34,35,36,37,38]. In addition, cohort studies in Nepal have found that TB stigma and discrimination in communities, hospitals, households, and especially self-stigmatization by people with TB, is prevalent [39,40,41]. 

### 2.2. Ongoing Studies Addressing the Socioeconomic Impact of TB in Nepal

Although no nationally representative TB patient costs survey has yet been completed in Nepal—a national survey is planned for 2021 with support from WHO—there have been two major longitudinal cohort studies using adapted WHO TB Patient Costs Survey methods administered at multiple time points during TB treatment. Both studies were implemented by Birat Nepal Medical Trust (BNMT), the first conducted during a TB-REACH Wave 5 Project and the second during the EU Horizon2020-funded IMPACT-TB Active Case Finding Project (www.impacttbproject.org). The findings of both studies were presented by BNMT team members at the workshop [42]. 

The surveys measured the economic impact of TB on affected households through measurements of: direct costs (out-of-pocket medical expenses, such as medicines and clinics; and non-medical expenses, including travel and accommodation related to clinics and additional food expenditure); and indirect costs (including lost time and income, opportunity costs, and coping strategies) [43]. The results showed that the economic burden on TB-affected households was high, especially due to costs relating to travel, lost income, and associated with seeking a mixture of both public and private healthcare. The prevalence of catastrophic costs of TB-affected households (total TB associated costs of more than 20% of the same TB-affected household’s annual income) [13,43,44] was high in both studies (53% and 49%, respectively) [42]. 

Other key findings of the studies were high prevalence of food insecurity amongst TB-affected households, and access to and engagement with TB diagnosis and care being associated with worsening poverty [42]. Active case finding (ACF) was able to mitigate some of these costs, especially those incurred pre-treatment, but was not able to fully eliminate catastrophic costs or reduce poverty or food insecurity. These results indicate that ACF alone will be insufficient to achieve elimination of catastrophic costs in line with the WHO 2015 End TB Strategy and suggest that there may be synergistic benefit from ACF being combined with integrated and comprehensive socioeconomic support and social protection packages [3,42]. 

In addition to the economic impact of TB, the psychosocial impact can also be pernicious and far-reaching, with studies from diverse country settings reporting high levels of stigma, discrimination, isolation, and mental illness, in people with TB [11,39,45,46,47,48,49]. While the TB Patient Cost Surveys described above were able to measure the economic impact to people with TB and their households, they were unable to shed light on the psychosocial impact of TB. Nor did these studies offer insight into suitable designs of integrated socioeconomic support interventions that could address both the psychosocial and economic implications of TB disease in Nepal. 

Since March 2018, a Wellcome Trust funded research project (209075/Z/17/Z), nested within the IMPACT-TB project, has been implemented in four districts of Nepal (Makwanpur, Chitwan, Dhanusha and Mahottari) to generate new local evidence to fill this knowledge gap. The mixed-methods research consisted of complementary studies: a cohort study of 221 people with TB recruited to the IMPACT-TB study (half of whom were found by ACF) with a cross-sectional comparator group of 120 TB-unaffected controls, selected by convenience sampling from communities within the same study sites, to evaluate the socioeconomic determinants and consequences of TB in Nepal; and a qualitative study gathering opinions of key stakeholders on the barriers and facilitators to accessing and engaging with TB diagnosis and care in Nepal. The same stakeholders were subsequently invited to the final phase of the research, the national workshop reported here, at which the preliminary findings of the cohort and qualitative study were disseminated. The primary aim of the workshop was to create a shortlist of feasible, locally appropriate socioeconomic interventions for TB-affected households in Nepal, which have the potential for further evaluation for scale-up through randomized controlled trials. 

Preliminary analyses of the cohort study data, presented during the workshop, showed that the social determinants and consequences of TB in Nepal were consistent with those found in diverse settings. After adjusting for age and sex, people with TB were more likely than TB-unaffected controls to have lower education levels, be unemployed, have greater food insecurity, use non-renewable smoke-producing fuels to cook, and perceive that they did not have enough money to meet their needs. Despite people with TB reporting high rates of trust and belonging in their local communities, one third of people with TB reported depression and/or anxiety, which was significantly higher than reported rates in TB-unaffected controls. This depression and anxiety may be compounded by stigma, which was reported by people with TB as prevalent, both in terms of self-stigmatization and enacted stigma experienced in the community. 

The qualitative study consisted of seven focus group discussions (FGDs) with 54 purposively selected key stakeholder participants. The stakeholders were pragmatically categorized into three groups: people affected by TB (consisting of 14 people affected by drug-sensitive TB, and seven affected by drug-resistant TB), community stakeholders (consisting of community leaders and elders, and representatives of grass-roots civil society organizations), and TB healthcare professionals (consisting of National TB Program (NTP) managers, physicians, community health workers, and volunteers who were associated with the NTP). Initial analyses of FGD discussions found that they focused on three major themes related to barriers and facilitators to TB diagnosis and care: socioeconomic condition, access to healthcare, and provision of healthcare. Within these themes, the identified barriers for people with TB in Nepal included economic hardship associated with TB illness, diagnostic delays (especially relating to patient pathways consisting of both public and private healthcare seeking), anxiety, and stigmatizing behavior from healthcare workers (Figure 1). 

The existing or required facilitators identified to overcome these barriers included integrated socioeconomic support, mass TB awareness programs, and patient-centered care. It was also recognized that strong political commitment, sufficient funding, and advocacy were required as underpinning factors to achieve the aims of these facilitators (Figure 2).

### 2.3. Section 1 of the Workshop

Section 1 of the workshop opened with presentations from the BNMT team highlighting the literature review and findings of the studies summarised above. This led on to a presentation by the Chief of the Quality, Standard and Regulation Division of the Ministry of Health and Population, highlighted the advances that Nepal has made towards improving access to healthcare and reducing the financial burden on patients and their households, through the implementation of social protection [43]. Nepal’s Social Security Act of 2017 legislated for a free package of basic healthcare and was expanded to cover more illnesses in 2018 through National Health Insurance regulation [50]. Social protection, including cash transfers, has been provided for some years to certain groups in Nepal (including old age allowances, child grants, and a recent, small-scale pilot scheme of grants for the ultra-poor) [51,52], and following humanitarian crises such as the 2015 earthquake [53]. 

During workshop discussions following the presentation, it was noted that, in order to achieve expansion of social protection coverage, increased funding and resources would need to be ring-fenced, across health, social care, and financial departments and sectors, political will would need to be mobilised, and that complementary community knowledge and awareness strengthening activities regarding TB would be essential. As one participant who leads a hostel for people with multi-drug resistant TB (MDR-TB) in Nepal suggested, “We need to open our eyes to social protection for TB-affected people, which must be effective, cost-effective, and evidence-based”. 

Workshop participants then reviewed and discussed the current economic support provided to people with TB in Nepal. It was noted that NTP-provided economic support predominantly focuses on ensuring access to hostels and covering nutritional and travel costs for people with drug resistant TB (DR-TB). This economic support amounts to Nepalese Rs. 3000 (approximately USD 26) per month for ambulatory patient and Rs. 1000 (~USD 9) per month for hostel-based patients throughout treatment. Workshop participants reported that the original impetus for this scheme was a perception that people with DR-TB in Nepal face the greatest psychosocial and economic challenges in accessing and engaging with care. Economic support is not routinely provided to people with drug sensitive TB (DS-TB) but, in late 2018, the government of Nepal committed to initiate a social protection programme for people with TB and HIV in Nepal. It was noted during workshop discussions that the current economic support has received limited impact evaluation and there was agreement that expansion of coverage to include other vulnerable people with TB should be considered [34]. 

### 2.4. Section 2 of the Workshop

#### (i) Identifying the existing psychosocial and economic support available for TB-affected households in Nepal

Participants were purposively split into six, small multi-disciplinary groups to discuss: the existing NTP-led psychosocial and economic support interventions for TB-affected households in Nepal; the advantages and disadvantages of each (including intervention design, logistics of delivery and implementation, funding and sustainability, acceptability to TB-affected households and impact if known); what refinements or improvements could be made to the existing intervention; and what related, new, potential interventions could be implemented and by whom such interventions would be delivered and funded. Each group was randomly assigned a specific discussion area: Groups 1 and 2 discussed the psychosocial element of interventions; and Groups 3 and 4, the economic element. Groups 5 and 6 discussed issues surrounding to whom such support is or should be targeted (e.g., all TB-affected households, only those with drug-resistant TB, only those in extreme poverty or with other features of vulnerability). The workshop agenda and group work materials can be found in the Appendix A.

Groups 1 to 4 identified that TB medication adherence counselling existed in Nepal, as did cash transfers for food and nutritional support, solely for patients with DR-TB (Table 1). The groups suggested that psychosocial counselling should also be offered that goes beyond existing medication adherence counselling and which could be integrated with counselling packages for other illnesses (e.g., diabetes). Additionally, it was agreed that mutual peer support groups would be beneficial (especially if led by TB survivors). Finally, the groups recommended that current cash transfer amounts were insufficient and should be increased, and coverage extended to other vulnerable TB-affected people (Table 1). 

#### (ii) Targeting of the intervention and potential funding sources

Groups 5 and 6 noted that a sizeable proportion of people with both DS- and, to a greater extent, DR-TB, have significant side effects from medications (including for example nausea and vomiting, diarrhea, or leg pains from peripheral neuropathy) but had to pay additional fees for medications to control these side-effects. Therefore, people with side effects were identified as a potential target recipient group. Groups 5 and 6 were unable to resolve a debate concerning whether support packages should be implemented using a blanket “one-size-fits-all” approach or a stratified, needs-based approach based on vulnerability as defined by eligibility for existing social protection schemes (Table 2). Concerns were raised about stratified approaches being logistically difficult to deliver and creating risk of misclassification of households (e.g., households classified as not vulnerable and therefore ineligible to receive support which are vulnerable and would benefit from receiving support).

All groups acknowledged that any increased funding would have to be co-funded between the NTP and NGOs, external donors (including, but not limited to, the Global Fund, USAID, Damien Foundation, TB-REACH, SAARC), or ideally across other governmental departments and ministries. Concerns were raised about the initial level of funding required and whether this would be sustainable in the long-term. It was also noted that, in the case of Nepal, the funding would have to be mobilized and coordinated across the national, provincial, and local levels.

#### (iii) Designing and voting on the most suitable interventions for evaluation in a trial

Based on the mixed-methods research findings, workshop discussions, and feedback from the group work, two potential outline options were generated for the psychosocial and economic elements of an intervention that could be evaluated as part of a trial in Nepal (Table 3). The psychosocial element included a community-based DOT provider or community health volunteer (CHV) being trained to assess stigma level during a household visit and then inviting the household to participate in either a peer-led mutual support group at district level [36] or a psychosocial counselling session about TB-related stigma and how to overcome it. Both the support group and counselling session would include education about citizens’ and patients’ rights, smoking cessation, indoor air pollution avoidance, and TB treatment and prevention. It was noted during the workshop that community-based DOT would have the complementary impact of partially mitigating costs and time associated with travel to DOTS clinics. Overall, 79% (30/38) opted for the household-level educational and psychosocial counselling session. The economic elements included a monthly basic cash transfer for all patients and people with DR-TB in hostel (at amounts above existing transfers, see Table 3) plus either conditional cash transfers for medication side effects arising during treatment or additional cash transfers for those TB-affected people or households deemed high-risk or vulnerable by the program. Definition of what constitutes high-risk was debated and not agreed upon, but would be likely to include elderly patients, single female patients, those living in extreme poverty, the homeless or people in unstable housing, and people unemployed. A total of 30/34 (88%) voted in favor of additional cash transfers for high-risk TB-affected people or households (Table 3).

## 3. Recommendations and Conclusions

This national workshop was the first of its kind in Nepal to review the evidence concerning the socioeconomic determinants and consequences of TB and to design a socioeconomic intervention package for TB-affected households. The workshop brought together key stakeholders from different sectors, including people with TB and DR-TB, and reviewed the latest local evidence on the socioeconomic impact, barriers, and facilitators of accessing and engaging with TB diagnosis and care in Nepal. The presentations and discussions during the workshop elicited that TB-affected people in Nepal face severe financial, social, mental, geographical, and health system level challenges. More broadly, it was repeatedly noted that the current low coverage of social protection in Nepal is likely to increase households’ vulnerability to poor health and worsening impoverishment, thus reinforcing the vicious cycle of TB and poverty. To address this, the participants agreed that comprehensive social and economic support was required for TB-affected households, which builds on the limited existing platforms. The wider recommendations arising from the workshop are summarized in Box 1.

Box 1Five key recommendations suggested during the workshop.The workshop recommended comprehensive support strategies, including psychosocial support and economic support, for people with TB.
***1.*** ***Psychosocial support:*** Psychosocial counseling should be incorporated into existing NTP activities to improve mental health of all people with TB. Counseling by health workers or community volunteers at the point of TB care, OPD visits in public and private health facilities, and/or during community-based DOT and household visits, could support people with TB to cope with the psychological impact of the disease, and potentially improve TB treatment adherence and outcomes. A more detailed evaluation and assessment of mental health of people with TB is required.***2.*** ***Economic support:*** To reduce the financial burden of TB, the current cash transfer amount for people with DR-TB should be increased and this economic support extended to people with DS-TB, especially the most vulnerable (e.g., the most impoverished, unemployed). It was also recommended that additional budget be provided to healthcare centers to subsidize provision of medication to treat people experiencing adverse effects of TB medication or requiring related ancillary services. A national TB Patient Costs Survey would provide more accurate estimates of out-of-pocket and lost income costs incurred during TB treatment, which would support related policy translation.***3.*** ***Social protection:*** Coverage of social protection packages should be expanded to other vulnerable populations that are defined by national social health insurance program to reduce the likelihood of developing TB disease.***4.*** ***Nutritional support:*** There is a need for nutritional support not only for people with MDR-TB but also for people with DS-TB. This recommendation was seen to align with the evidence presented during the workshop that people with TB, regardless of drug-resistant status, experienced high levels of food insecurity.***5.*** ***Education and public-private mix (PPM):*** A PPM approach is vital to avoid the unnecessary expenditure incurred at private health care visits for diagnostic and treatment services that are free in the public healthcare system. Educational campaigns to inform communities about TB signs, symptoms, and free NTP TB diagnostic and treatment services would complement this approach. A PPM strategy has recently been approved by the Ministry of Health and Population in June 2019 and will be crucial to make TB diagnosis and treatment affordable.

The overarching aim of the workshop, to develop a shortlist of socioeconomic interventions for TB-affected households to be evaluated in a randomized controlled trial, was achieved. The preferred, chosen intervention integrated: (i) psychosocial counselling by a community-based DOT provider or volunteer during a household visit to reduce stigma and increase TB-related knowledge; and an enhancement of the amount of the existing nutritional and transport economic support with expansion of the support coverage to more vulnerable TB-affected people. The evaluation of socioeconomic vulnerability of TB-affected households and the design of the intervention are being refined in conjunction with the NTP and multi-sectoral partners for future implementation and robust evaluation in a large, randomized controlled trial.

## Figures and Tables

**Figure 1 tropicalmed-05-00098-f001:**
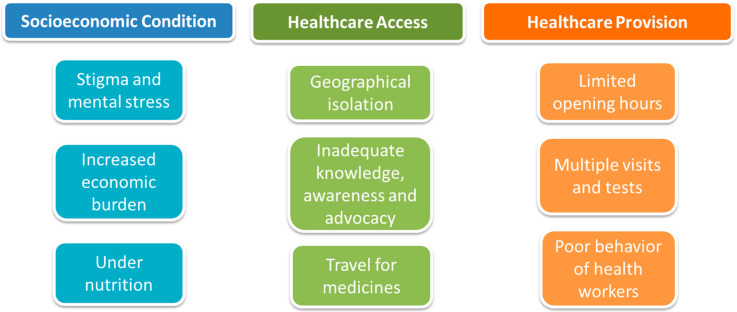
Perceived barriers to accessing and engaging with tuberculosis (TB) diagnosis and care.

**Figure 2 tropicalmed-05-00098-f002:**
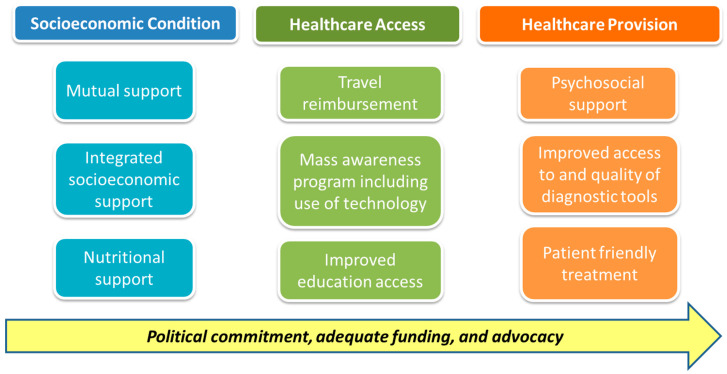
Perceived facilitators to accessing and engaging with TB diagnosis and care.

**Table 1 tropicalmed-05-00098-t001:** Current and potential psychosocial and economic interventions.

Intervention Element	Existing Interventions	Refinements of Existing Interventions	Suitable Potential Interventions	Mode of Delivery
**Psychosocial**	Routine NTP-led counseling for TB-affected people, which focuses solely on TB treatment adherence There is no routine counselling available to address the psychosocial impact of TB (e.g., stigma)	Existing counselling about adherence to TB medications could be supplemented by complementary psychosocial counselling	Educational and social awareness campaigns Provision of complementary counseling focused on stigma, low mood and depression, and isolation and marginalization Integrated psychosocial counseling for people with TB and comorbidities (e.g., diabetes, depression)	Psychosocial counseling to patients/households by trained health workers in healthcare facilities or in people’s homes Peer counseling and mutual support (“TB survivors group”) Research into psychosocial status of TB-affected people throughout treatment, especially vulnerable groups
**Economic**	Rs 3000/month as nutritional/transport allowance for ambulatory DR cases Rs 1000/month for hostel-based DR cases Community-based DOT: Rs 1500 for volunteers, Rs 500 for health worker	**For people with DS-TB** Rs 1500/month nutritional allowance for every DS case Rs 500/month transport allowance (conditional) for DOT and follow up Rs 1000 for side effect management (conditional) **For people with DR-TB** Rs 3000/month for transportation Rs 2000/month for nutrition Rs 5000 for side effect management (conditional)	Income generation activities or back-to-work schemes Cash transfer schemes Incentivized TB screening program (e.g., economic, nutritional, or other incentive for attending TB screening facilities, being screened for TB if appropriate, and/or having a positive TB smear or culture) Nutritional package program for TB-affected person and their household	Conditional cash transfer through bank accounts or mobile phones Household visits to reduce travel costs and support TB treatment and prevention activities Partnering with local organizations to provide support and DOT Community sensitization and mobilization Microcredit and vocational training

**Table 2 tropicalmed-05-00098-t002:** Definitions of vulnerable groups and benefits and drawbacks of targeted support.

Vulnerable Groups	Targeting Support	Benefits	Drawbacks
Economic status (defined by eligibility for existing social health insurance programs) Limited geographical access to healthcare Co-morbidities (e.g., HIV, diabetes) Other specific groups including: children, elderly population, pregnant women, separated men and women, ethnic minorities, people with disabilities, homeless, daily waged workers or unemployed	Two main options for provision of socioeconomic support were discussed: a blanket “one size fits all” approach given that the majority of TB-affected households are economically and socially deprived support stratified by estimated socioeconomic vulnerability defined by meeting eligibility criteria for existing social health insurance and/or meeting other definition of belonging to a vulnerable group	Needs-based socioeconomic support Address both socioeconomic determinants and consequences of tuberculosis Enhance early diagnosis and prompt treatment and potentially interrupt transmission Improve TB case notification, treatment adherence and outcome A rights-based approach would ensure fundamental human rights for health were met Reduces stigma and discrimination as well as awareness Mitigation of catastrophic costs by defraying both direct and indirect TB-related costs (e.g., lost income)	Chances of bias and mis-categorization of TB-affected households as vulnerable or not with stratified support Concerns were raised concerning the potential for creation of dependency on support beyond TB illness among the person with TB and their household Huge economic burden to the country and health system of implementing and scaling up socioeconomic support Financial feasibility and sustainability would depend on start-up and maintenance costs of support scheme and funding stream (e.g. burden on National TB Program or costs shared across governmental departments including, for example, those related to health, social inclusion, and job security) Concerns were raised concerning accountability and transparency of the program

**Table 3 tropicalmed-05-00098-t003:** Design of and votes for psychosocial and economic elements of potential integrated socioeconomic support package for people with TB.

Psychosocial Package	Votes	Economic Package	Votes
**A.** CB-DOT provider / CHV assesses stigma level at household visit (or in hostel) and invite whole household to local mutual peer support TB group	**21% (8/38)**	**A. Economic intervention:** Monthly basic cash transfer for all patients (DR-TB NRs 6000; DS-TB NRs 1000)* Pocket money for people with DR-TB in hostel NRs 1500 Conditional cash transfer for medication side effects to health centre (DR-TB 5000 NRs; DS-TB 1000 NRs.)	**12% (4/34)**
**B.** CB-DOT provider / CHV assesses stigma level at household visit and do stigma counselling session with household during visit	**79% (30/38)**	**B. Economic intervention:** Monthly basic cash transfer for all patients (DR-TB NRs 6000; DS-TB NRs 1000)* Pocket money for people with DR-TB in hostel NRs 1500 Additional cash transfer for socially high-risk persons (to be defined) with TB (3000 NRs.)	**88% (30/34)**

Legend: The anonymized vote took place at the end of the workshop. Thirty-eight participants were present to cast votes. The option of not casting a vote was given and four votes were not cast for the economic element of the intervention. *The sections of the cells that are underlined show the differences between psychosocial packages A and B*, *and economic packages A and B*. *The sections of text not underlined in these cells show the elements which are the same in both psychosocial packages A and B, and economic packages A and B*.

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
