# Peer review of "Developing Feasible, Locally Appropriate Socioeconomic Support for TB-Affected Households in Nepal"

_tropicalmed, 2020, doi:10.3390/tropicalmed5020098_

Round 1

Reviewer 1 Report

The authors described the activities and outcomes arising from the national workshop. This manuscript is meaningful since it explored several policy options to mitigate the financial burden of TB patients in Nepal. However, I would like to raise some concerns about the manuscript.

*Major

  1. The format of the manuscript included both the literature review and the workshop. Even though their contents are linked, the format looks confusing. The methods of the literature review are not clearly elaborated and the reason the manuscript combines workshop report with a literature review is not very clear.

*Minor

1. page 1 line 41-43: This quotation does not look essential.

Reviewer 2 Report

This paper reports on a workshop on the question how to introduce in Nepal stronger psycho-socio-economic support for TB patients. This is an important topic; the end TB strategy has as one of its targets to have 0% catastrophic costs by 2020. Urgent action is needed in this area!

There are a few issued (see below), indicating that MINOR REVISION is needed.  

A) Line 56. We currently have: “a target of “zero TB-affected families facing catastrophic costs” by 2030” But the target is to have this by 2020 already! Actually, missing this overly ambitious target should be discussed. Also, I think a few studies that have actually assessed the percentage of TB families with catastrophic costs should be mentioned (e.g. by Viney et al, 2019: The Financial Burden of Tuberculosis for Patients in the Western-Pacific Region).

B) The expression “DOTS” is often used. However, sometimes, this should probably be DOT, i.e. directly observed therapy, e.g. in Table 1: “allowance for DOTS and follow-up” (and also “…to provide support and DOTS”). You may actually want to put an asterisk to explain. The area of treatment supervision and support warrant some discussion: if DOT is home-based, then cash for DOT is not really needed. If DOT is strictly health-facility based, then a large cash support is needed.

C) Line 271: It states that patients with side effects may receive additional cash support. Somewhere else (recommendation 2) you mention that ancillary treatment should be offered for free. I believe that really, ancillary treatment should be provided for free, rather than organising cash support for patients with side effects.  

D) Table 1: I do not understand the following: “Incentivized TB screening program”.   

Details:

 1.     The references need to be written down in accordance with the rules of the journal.

2.     Reference 47 lacks the year.

3.     Reference 8 and 13 are identical

4.     I would drop the workshop photos.  

5.     Line 301: I would NOT put in brackets the issue of debate.

6.     To safe space you may want to provide a leaner version of the AGENDA (e.g drop the column Responsible and drop rows like registration, coffee breaks etc).

7.     Lines 345-346 are coloured yellow. (there may be a reason for this that has not yet been addressed).

Round 2

Reviewer 1 Report

I accept the present form of the manuscript.